# Effect of Rough Surface Platforms on the Mucosal Attachment and the Marginal Bone Loss of Implants: A Dog Study

**DOI:** 10.3390/ma13030802

**Published:** 2020-02-10

**Authors:** Javier Montero, Alberto Fernández-Ruiz, Beatriz Pardal-Peláez, Alvaro Jiménez-Guerra, Eugenio Velasco-Ortega, Ana I. Nicolás-Silvente, Loreto Monsalve-Guil

**Affiliations:** 1School of Dentistry, Faculty of Medicine, University of Salamanca, 37007 Salamanca, Spain; javimont@usal.es (J.M.); bpardal91@gmail.com (B.P.-P.); 2Private Practice Specialized in Oral Surgery and Implant Dentistry, 07800 Ibiza, Spain; bertoynacho@hotmail.com; 3Faculty of Dentistry, University of Seville, 41009 Seville, Spain; alopajanosas@hotmail.com (A.J.-G.); lomonsalve@hotmail.es (L.M.-G.); 4School of Dentistry, University of Murcia, 30008 Murcia, Spain; ainicolas@um.es

**Keywords:** implant platform, mucosal attachment, marginal bone loss

## Abstract

The preservation of peri-implant tissues is an important factor for implant success. This study aimed to assess the influence of the surface features of a butt-joint platform on soft-tissue attachment and bone resorption after immediate or delayed implant placement. All premolars and first molars of eight Beagle dogs were extracted on one mandible side. Twelve-weeks later, the same surgery was developed on the other side. Five implants with different platform surface configurations were randomly inserted into the post-extracted-sockets. On the healed side, the same five different implants were randomly placed. Implants were inserted 1 mm subcrestal to the buccal bony plate and were connected to abutments. The primary outcome variables were the supracrestal soft tissue (SST) adaptation and the bone resorption related to the implant shoulder. The SST height was significantly larger in immediate implants (IC95% 3.9–4.9 mm) compared to delayed implants (IC95% 3.1–3.5 mm). Marginal bone loss tended to be higher in immediate implants (IC95% 0.4–0.9 mm) than in delayed implants (IC95% 0.3–0.8 mm). Linear-regression analysis suggested that the SST height was significantly affected by the configuration of the platform (0.3–1.9 mm). Roughened surface platforms resulted in higher SST height when compared to machined surface platforms. Marginal bone loss was less pronounced in roughened designs.

## 1. Introduction

The biological phases of the healing dynamics of both hard and soft tissues after dental implant placement have been reported in animal experiments [1,2] and human studies [3]. Such dynamics have also been described for implants immediately placed into extraction sockets [4,5,6].

Marginal bone resorption occurs when an implant is inserted both in healed sites [7,8] and also in extraction sockets [4,7]. A systematic review reported that marginal bone resorption takes place irrespectively on the implant neck configuration [9]. Nevertheless, roughness and micro-threads at the implant neck contribute to marginal bone preservation [10,11,12,13,14]. Few studies have compared the effect of implant surface configurations and geometries on the peri-implant healing process at implants placed immediately into extraction sockets [15,16]. It seems that the presence of marginal micro-threads or grooves reduces marginal bone resorption and soft tissue recession. To date, most research regarding the clinical management of immediate implants is mainly focused on hard tissue integration and the effect of different implant designs or surgical approaches on clinical esthetic outcomes. However, there is insufficient evidence whether or not different implant designs and surface configurations enable the formation of an adequate soft-tissue seal at implants immediately placed into extraction sockets and compared to implants placed into fully-healed alveolar ridges.

The objective of this experimental study was; therefore, to assess the influence of the surface roughness platform of butt-joint implants placed immediately into extraction sockets, with regards to the dimensions of the supracrestal peri-implant soft tissue seal and the preservation of marginal alveolar bone 12 weeks after implant placement. 

## 2. Materials and Methods 

The experimental protocol was submitted and approved by the Ethical Committee of the University of Santiago of Compostela in Spain (Ref: 01/16-LU/001). The study was reported according to the ARRIVE guidelines.

### 2.1. Animal Sample

The guidelines for animal care adopted in Spain were strictly followed. Eight 3-year-old male Beagle dogs supplied by Isoquimen SL (Sant Feliú de Codines, Barcelona, Spain; RN° B9900039) were included in the study. To minimize the number of dogs for ethical reasons (Three R requirements), but to maintain enough power of analysis, a split-mouth design was developed. The dogs were the unit of analysis. The sample size was estimated to be eight dogs for detecting differences in the supracrestal soft tissue height of 2 mm with a power of 80% and a probability of alpha = 0.05. 

### 2.2. Randomization and Allocation Concealment

The randomization for the side of the mandible and the allocation of five different types of implants were performed by one of the authors that did not participate in the surgeries (AF). Sealed envelopes were used and opened at the time of surgery. The surgeon (EV) was informed about the selection of the hemimandible for implant placement before the first session of tooth extraction.

The histological slides did not report indications about treatment allocation, nor did the examiner (JM) receive such information.

### 2.3. Surgical Preparations

The animals were pre-sedated with an intramuscular injection using a combination of 25 μg/kg medetomidine (Doctor Esteve, Barcelona, Spain) and 0.5 mg/kg morphine (Morfina Braun 2%, Braun, Barcelona, Spain). Then, general anesthesia was induced using 5 mg/kg/i.v. of propofol (Vetofol, Esteve, Barcelona, Spain) by intravenous infusion, followed by laryngeal intubation and the inhalation of a mix of 2.5%–4% isoflurane/sevoflurane and 30% oxygen from the ventilation tube. 

During the anesthesia, a veterinarian continuously recorded the electrocardiogram, arterial pressure, pulse oximetry, and capnography with non-invasive methods. Besides, submucous injections of articaine hydrochloride (Ultracain®, Normon, Spain) were infiltrated locally for bleeding control. After the intra-sulcular incisions, full-thickness mucoperiosteal flaps were made, and all mandibular premolars and first molars were extracted at one randomly selected hemi-arcade. The flaps were closed with single interrupted 4-0 vicryl resorbable sutures.

Twelve weeks after surgery, the same procedure was performed on the contralateral side, and five implants were installed into the extraction sockets in all premolars (P_1_–P_4_) and molar (M_1_) sites. Straight multi-unit screw-retained abutments were fixed with a closing screw onto all implants. Each multi-unit abutment was by the same manufacturer as its corresponding implant (Institute Straumann AG, Basel, Switzerland and Galimplant SLU, Sarria, Lugo, Spain).

In the healed contralateral ridge, five delayed implants were inserted in the healed sites of P_1_–P_4_ and M_1_. All implants received multi-unit abutments fixed with the closing screw. The implants were placed 1 mm subcrestal to the buccal plate level on both sides. Single 4-0 vicryl resorbable sutures were used to close the flaps, allowing non-submerged healing.

### 2.4. Implant Characteristics

In each hemiarcade, five different types of implants of two different manufacturers were assessed (Figure 1): Type I: Straumann BL (bone-level) implant with 3.3 mm diameter and 8 mm in length (Institute Straumann AG, Basel, Switzerland), with a body implant treated with SLA^®^ surface (mean R*a* value of 1.66 µm) and a machined platform (mean R*a* value of 0.27 µm).Type II: IPX-Std (standard) with 3.5 mm diameter and 8mm in length (Galimplant SLU, Sarria, Lugo, Spain), with an implant body and neck treated using a Nanoblast Plus^®^ technique (mean R*a* value of 1.69 µm) and a fully machined platform (mean R*a* value of 0.28 µm).Type III: IPX-Half, with 3.5 mm diameter and 8 mm in length (Galimplant SLU, Sarria, Lugo, Spain) with an implant body, neck and the outer half of the platform surface treated with Nanoblast Plus^®^ technique (same R*a* value described in Type II treated surface) and the inner half of the platform machined (same R*a* value described in Type II machined surface).Type IV: IPX-Full, with 3.5 mm diameter and 8 mm in length (Galimplant SLU, Sarria, Lugo, Spain) with the implant body, neck and the whole platform treated with the Nanoblast Plus^®^ technique (mean R*a* value of 1.69 µm).Type V: IPX-Control, with 3.5 mm diameter and 8 mm in length (Galimplant SLU, Sarria, Lugo, Spain), with implant body, neck, and platform machined (mean R*a* value of 0.28 µm).

The characteristics of each implant group are summarized in Table 1.

The IPX implant surface received a surface conditioning treatment consisting of a mixed method (mechanical and chemical; The Nanoblast Plus® technique) for improving implant surface topography through two consecutive processes: aluminum oxide spraying and triple acid etching. This method was patented by the manufacturer Galimplant SLU (Sarria, Lugo, Spain). The SLA (Sandblast, Large-Grit, Acid-Etching) method is a method patented by Straumann (Straumann AG, Basel, Switzerland) that induces surface erosion by applying a strong acid onto the blasted surface. This treatment combines blasting with large-grit sand particles and acid etching sequentially to obtain macro roughness and micro pits. The roughness characteristics of each surface can be observed with a magnification of 1000× in Figure 2.

Within each experimental group, half of the implants were placed in the healed ridge, and the counterparts were located into fresh extraction sockets. The IPX-Control implants were always inserted at the most distal site (M_1_), but the rest of the implants were randomly assigned using closed envelopes to distinct premolar sites (P_1_–P_4_). 

### 2.5. Maintenance Procedures

The protocol for postoperative care included a daily dose of cefovecin sodium (8 mg/kg/ Convenia®, Zoetis, Spain) and morphine (0.5 mg/Kg/day) injected intramuscularly during five days, as well as meloxicam (0.2 mg/Kg/p.o.), dispensed with meals. The sutures were removed one week post-surgery. Plaque control was performed by brushing the dogs’ teeth three times per week, with a 2% chlorhexidine gel. 

### 2.6. Euthanasia

Twelve weeks after implant placement, all animals were euthanized by an overdose of pentobarbital (40–60 mg/Kg/IV), and mandibular biopsies were obtained for histological processing.

### 2.7. Histological Preparation

The tissue blocks containing the implants and surrounding tissues were prepared for non-decalcified sectioning, where blocks were fixed in buffered 10% formaldehyde for at least four days at 4 °C. Afterward, the specimens were dehydrated in a series of ascending concentrations of ethanol solutions for 72 h at each stage (80%, 96%, 100%), and finally embedded in polymethylmethacrylate resin during three weeks at room temperature (until the resin was hard to the touch). Final curing was achieved by placing the samples into a 37 °C oven for 72 h. The blocks were then sectioned with a precision cut-off Exakt machine (Exakt Aparatebau GMBH, Norderstedt, Germany) and polished using silicon carbide grit papers in a descending size-grain protocol (from #600 to #4000 grit papers). Using this approach, 2 to 4 slices of 100–200 μm-thick sections were obtained buccolingually for the implants and subsequently analyzed. The specimens were stained using the Levai Laczko technique [17] (Figure 3 and Figure 4).

Photomicrographs were obtained using a Zeiss-Stemi 2000-C photomicroscope (Carl Zeiss Microimaging, Thornwood, NY, USA) at a magnification of 4×. An image analysis program (Image Pro Premier, Media Cybernetics, USA; CellSens, Olympus, Japan) was used to measure the peri-implant tissue healing based on the following landmarks: Implant shoulder (IS); the most coronal point of contact between bone and implant (B); the margin of the peri-implant mucosa (PM); and the apical border of the junctional epithelium (aJE).

These topographic references (Figure 5) were used to measure the following linear parameters (mm): The supracrestal soft tissue height (the distance between PM and B); the epithelial attachment length (the distance between PM and aJE); the connective adaptation heights (the distance between aJE and B); and the marginal bone loss (the distance between IS and B). In addition, the percentage of bone-to-implant contact (BIC) was measured in both the buccal and lingual sides of the implants. 

### 2.8. Data Analysis

Mean values and standard deviations (SD) were calculated for each outcome variable. Differences between the immediate and delayed implant sites were analyzed using Student’s t-test for paired observations. Differences between the distinct groups of implants were analyzed by ANOVA with post hoc Bonferroni corrections. Spearman correlation coefficients were used to evaluate the linear relationship between quantitative and ordinal variables. Finally, backward stepwise linear regression models were calculated for predicting both supracrestal soft tissue adaptation length and the average marginal bone loss. The level of significance was established at 0.05. All analyses were carried out with SPSS 18.0 (SPSS Inc., Chicago, IL, USA) software. Data were corrected with SUDAAN 11 Statistical Software (RTI International, SC, USA) for the autocorrelation of all variables derived from this split-mouth design study.

## 3. Results

All implants healed uneventfully without clinical signs of inflammation in the peri-implant mucosa. 

The supracrestal soft tissues were comprised of a dense connective tissue and a mature junctional epithelium. The connective tissue was rich in collagen, poor in vessels, and in some cases, with only a few scattered inflammatory cells at the level of the implant shoulder. Adjacent to this area, collagen fibers ran parallel to the long axis of the implant. Longer connective and epithelial layers were identified among immediate implants as compared to the delayed implants.

The supracrestal soft tissue height (formerly: biological width) (PM-B; Table 2) was significantly smaller at the lingual than at the buccal bony plate; This difference was statistically significant for both immediately placed and delayed implants (p ˂ 0.001). Moreover, the supracrestal soft tissue height was significantly larger for implants immediately placed into extraction sockets (IC95% 3.9–4.9 mm) than for the delayed placed implants (IC95% 3.1–3.5 mm). Intergroup comparisons using ANOVA showed that the supracrestal soft tissue height at the lingual side was significantly shorter for Strauman BL (1.8 ± 0.2 mm) implants than for IPX-Std (3.0 ± 0.5 mm) or IPX-Control (2.9 ± 1.1 mm) delayed implants.

Regarding the width of the epithelial barrier (Table 3), this was significantly shorter for delayed (IC95% 1.5–1.8 mm) than for immediately placed implants (IC95% 1.9–2.5 mm). No statistically significant differences resulted from the inter-group comparison by ANOVA.

The “connective tissue attachment” was significantly wider at immediately placed (IC95% 1.8–2.6 mm) than at delayed implants (IC95% 1.5–1.8 mm) (Table 4). Moreover, for delayed implants, the buccal connective tissue seal was significantly thinner for Straumann BL (1.1 ± 0.5 mm) than for IPX-half (2.0 ± 1.0 mm) implants.

The marginal bone loss (IS-B) tended to be higher for immediately placed (IC95% 0.4–0.9 mm) than for delayed implants (IC95% 0.3–0.8 mm), although the difference did not reach statistical significance (Table 5). The inter-group comparison by ANOVA was not significant either, but it was observed that the marginal bone loss was inversely correlated (r_s_ = −0.46; *p* = 0.008) with the extension of the modified platform (null: Strauman and IPX-Std; half: IPX-Half; Full: IPX-Full), but only for delayed implants.

The differences in BIC% between delayed implants (IC95% 60.6%–72.6%) and immediately placed implants (IC95% 53.1%–65.5%) was statistically significant (Table 6).

The results of the linear regression analyses (Table 7) show that the supracrestal soft tissue height around dental implants ranged, on average, from 2.9 to 4.1 mm. This height was significantly reduced (between 0.5–1.6 mm), as implants were placed into healed ridges, and significantly increased according to the extension of platform modification (0.3–1.9 mm), and compared to the healing of implants with non-modified platforms (IPX-Std and Straumann). Likewise, the average marginal bone loss ranged from 0.6 to 1.1 mm, and this loss could be minimized by 0.1–0.5 mm when the implants were inserted into healed bone or when the platform of the implants was modified (0.1–0.8 mm); in that, the greater the size of the area modified on the platform, the smaller the amount of bone loss. 

## 4. Discussion

This experimental study evaluated the influence of the surface roughness features at the implant platform on the peri-implant soft tissue and marginal bone loss, after 12 weeks of non-submerged healing, when implants were placed immediately or 12 weeks after tooth extraction. In the present study, three types of controls were used, the negative control (IPX-Control) providing an implant with a turned surface without modification (Figure 1). The positive control was a commercially available IPX implant (IPX-Std), with a non-modified platform, but an implant surface modified with a mechanochemical method (Nanoblast Plus®). The reference control was a Straumann BL implant, with the implant surface being modified with a different mechanochemical process (Sandblast, Large-Grit, Acid-Etching), but with a non-modified platform. The use of different types of IPX implants with the same dimensions (diameter and length), thread design, and surface treatment, but with different areas of roughness platforms, allowed us to separately analyze the influence of platform treatment on the formation of biological width and crestal bone remodeling. However, the data resulting from the machined-treated implants should be handled with caution, since the anatomical locations were not chosen randomly, being always placed at the M1 site. This was the reason for eliminating this group from the regression analyses and why the reference group was comprised of implants with a non-treated platform (IPX-Std and Straumann BL).

The rationale for modifying (roughening) the platform is based on findings from a report [18] that showed how bone tissue was able to grow over the micro gap, establishing the first bone-to-implant contact directly on the healing abutment using subcrestally placed implants. Moreover, the roughly modified surface provided optimal healing conditions by promoting coagulum adhesion and stability [15,19]. 

Several studies have focused on specific modifications of the implant neck and surface configurations to improve clinical performance of immediately placed implants [16,17].

The major finding in the present study is the fact that both the height of the supracrestal connective tissue and the length of the junctional epithelium were significantly greater at implants immediately placed in extraction sockets, compared to implants placed in healed sites. Moreover, the surface area of the modified roughened platform was proportional both to the length of the supracrestal soft tissue adaptation and also to the degree of marginal bone preservation.

Berglundh et al. [2] stated that the peri-implant gingival tissues of delayed implants (placed into healed ridges at the crestal level) measure 3.5 mm (2 mm epithelium + 1.5 mm connective) and need at least six weeks to establish a soft tissue barrier with the proper dimensions and tissue organization. But the values mentioned above may change when implants are placed immediately after tooth extraction and/or at the subcrestal position. Our histometric analysis showed that the biological width was larger among immediate (IC95% 3.9–4.9 mm) implants than among deferred implants (IC95% 3.1–3.5 mm) 12 weeks after implant placement. This is in agreement with the findings of some reports [6,17], although in this study the magnitude of the biological width was found to be larger than that reported by de Sanctis et al. [17] concerning four commercially available implant systems (i.e., 3i Osseotite Certain; Astra MicroThreadt-OsseoSpeedt; Thommen SPI Element^s^; and Straumann ITI standard) placed immediately into the distal socket of the third and fourth premolar of eight beagle dogs. De Sanctis et al. found that the biological width, six weeks after implant placement, ranged between 3.4–4.2 mm and 2.9–3.2 mm at the buccal and lingual aspects, respectively. Moreover, the experimental conditions tested in this study (IPX-Full and IPX-Half) afford even thicker mucosal attachments. These discrepancies could be based on the fact that in the Sanctis study [17] the implant shoulder was placed at the level of the marginal portion of the buccal plate for 3i and Astra implants, or supracrestally for Straumann and Thommen implants, which have a polished collar of 1.8 mm. In this study, all bone-type implants were placed 1mm subcrestally, generating a deeper mucosal seal, which has also been reported elsewhere [19].

The fact that the supracrestal soft tissue adaptation at immediately placed implants was significantly longer than that at delayed implants is in agreement with the findings from other studies [6,15,17,20]. 

Some authors have found that biological width and marginal bone loss are more or less identical when using either one-piece or two-piece implant configurations [21]. It was also shown that the most pronounced loss of crestal bone and the most severe degree of peri-implant inflammation occurred when the interface (micro gap) was placed 1 mm below the crest [22]. The lowest bone loss and the least extension of inflammation occurred adjacent to two-piece implants when their micro gap (interface) was located exactly at the level of the bony crest. Additionally, the least amount of bone resorption/peri-implant inflammation was observed when the micro gap (interface) was located 1 mm above the alveolar crest [22]. Likewise, it was confirmed that the length of the junctional epithelial and the height of the supracrestal soft tissue adaptation were significantly greater for implants placed 1 mm below the alveolar crest, compared to implants placed either flush with the level or 1 mm above the alveolar crest [23].

In the present study, the soft tissue dimensions around both immediately placed and delayed placed implants tended to be greater at the buccal compared to the lingual aspects. This is in contrast to reports from other authors [6,17]. This may be related to the thickness of the residual bony plates after mechanical instrumentation, like extractions and drilling [24]. Bone resorption is even more extensive around immediately placed implants placed in multiple adjacent tooth extraction sites, as carried out in the present study, as compared to immediately placed implants into single extraction sites [25,26]. Hence, it may be hypothesized that the extent of bone resorption may be due to the underlying thickness of the peri-implant soft tissues at the time of implant installation, as established in dog studies [27,28,29].

Another reason for the longer soft tissue dimensions observed in the present study could be due to the elevation of full-thickness flaps before implant placements. In an experiment in dogs, ridge alterations following immediate implant placement with and without flap surgeries were studied [30]. A longer supracrestal soft tissue compartment was found in the group of animals where a flap had been performed. An explanation might be that the mobilization of the mucoperiosteal flap impaired the vascular supply for the healing site during the initial phase [31].

Regarding the effect on marginal bone loss, the experimental implants tested in the present study seem to have contributed to the preservation of the alveolar bone. Similar results were observed in another experiment in dogs, in which a moderate marginal bone loss of approximately 0.6 mm around the immediate implants and a long soft tissue seal of 4.8 mm was reported [6].

The order of magnitude of the differences in the length of the junctional epithelium and the supracrestal connective tissue seal in the two implant modalities (immediate and delayed) were modest in the present study. Although statistically significant, the clinical implications of these differences for implant longevity remain to be explored.

The present study possessed some strength as a strict development of the protocol but also has some limitations. Future research should address the stability of these larger epithelial-connective barriers and their ability to ensure the mucosal seal.

## 5. Conclusions

Within the limitations of the present study, it may be concluded that the effect of different topographical platform designs on marginal bone levels and soft-tissue height is inconclusive. Rough platform modifications may be beneficial to better preserve the marginal bone level in comparison to non-modified turned platforms. Regarding the timing for placing the implant, it seems that better results were obtained with delayed implants than with immediate implants.

## Figures and Tables

**Figure 1 materials-13-00802-f001:**
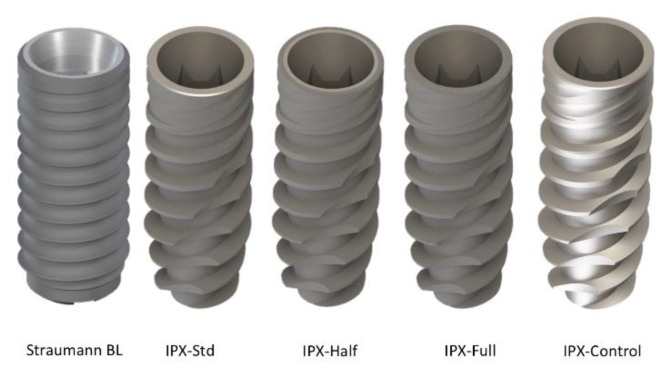
The topographic profiles and platform surfaces of the five implant designs evaluated: Straumann BL platform with SLA^®^ surface platform; IPX-Std with a fully machined platform; IPX-Half with Nanoblast Plus^®^ surface in the outer half of the platform and machined surface in the inner half of the platform; IPX-Full with the whole platform treated with Nanoblast Plus^®^ and IPX-Control with full body and platform machined.

**Figure 2 materials-13-00802-f002:**
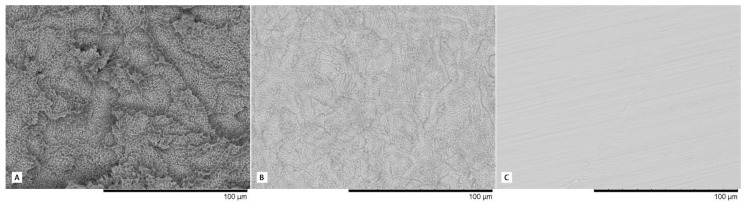
High magnification images (1000×) of the surfaces evaluated; (**A**) SLA^®^ Surface, (**B**) Nanoblast Plus^®^ Surface, and (**C**) IPX Machined Surface.

**Figure 3 materials-13-00802-f003:**
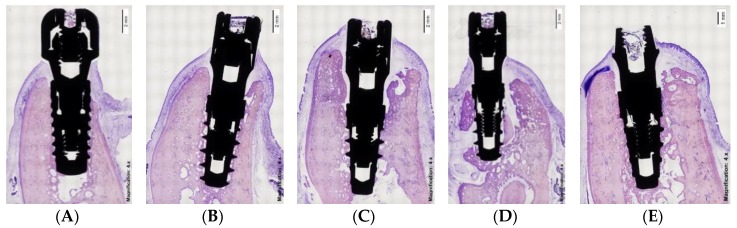
Histological images of immediate implants. (**A**) Straumann BL; (**B**) IPX-Std; (**C**) IPX-Half; (**D**) IPX-Full; (**E**) IPX-Control.

**Figure 4 materials-13-00802-f004:**
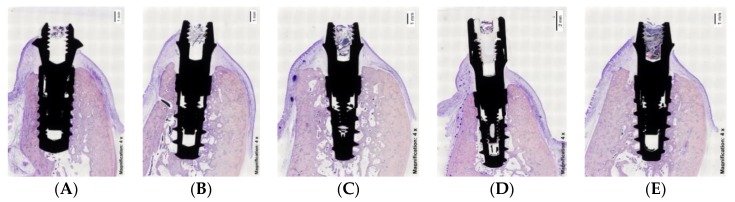
Histological images of delayed implants. (**A**) Straumann BL; (**B**) IPX-Std; (**C**) IPX-Half; (**D**) IPX-Full; (**E**) IPX-Control.

**Figure 5 materials-13-00802-f005:**
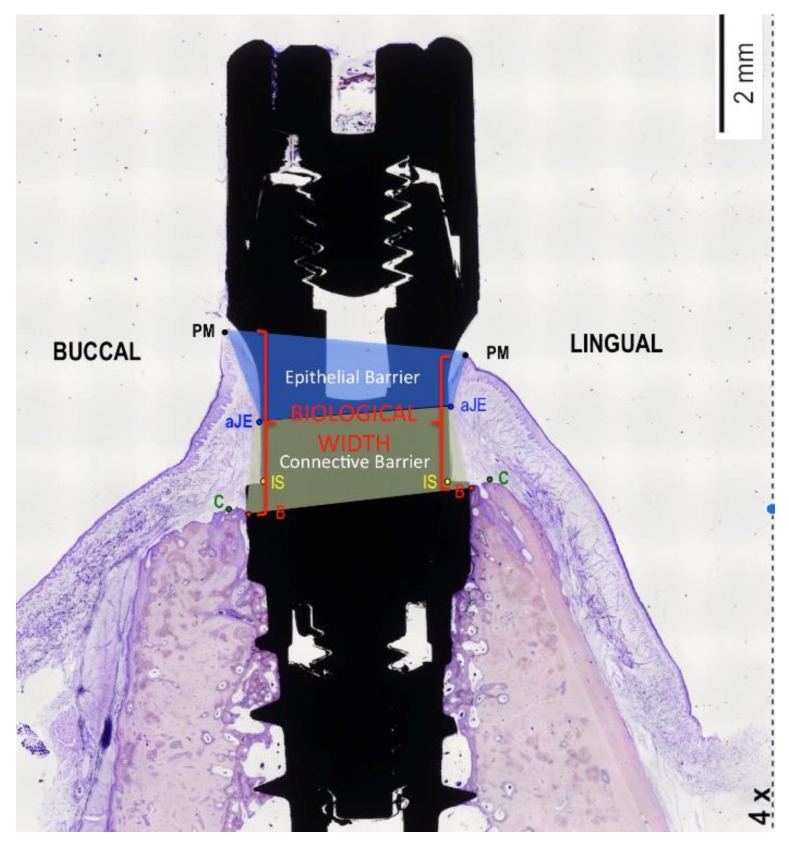
Diagram illustrating the landmarks for histological evaluation. IS—implant shoulder; B—most coronal bone-to-implant contact location; C—top of the alveolar crest; aJE—the apical border of the junctional epithelium; PM—the top of the margin of the peri-implant mucosa.

**Table 1 materials-13-00802-t001:** Summary of the main characteristics of each experimental group, including implant design, implant size, body treatment, platform treatment, and R*_a_* values (mean and standard deviation) for both treated and machined surfaces.

Experimental Group	Implant Design	Implant Size	Body Treatment	Platform Treatment	R*_a_* Value Treated Surface	R*_a_* Value Machined Surface
Straumann BL	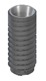	3.3 × 8	SLA^®^ Surface	Machined	1.66 µm	0.27 µm
IPX-Std	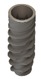	3.5 × 8	Nanoblast Plus^®^ Surface	Machined	1.69 µm	0.28 µm
IPX-Half	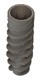	3.5 × 8	Nanoblast Plus^®^ Surface	Outer Half: Nanoblast Plus^®^ SurfaceInner Half: Machined	1.69 µm	0.28 µm
IPX-Full	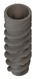	3.5 × 8	Nanoblast Plus^®^ Surface	Nanoblast Plus^®^ Surface	1.69 µm	Not Apply
IPX-Control	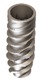	3.5 × 8	Machined	Machined	Not Apply	0.28 µm

**Table 2 materials-13-00802-t002:** Comparison of the biological width (the distance between PM and B) between deferred and immediate implants in both the buccal and lingual sides (mm) among the implant subgroups. * Statistically significant inter-group comparisons according to ANOVA test (*p* < 0.05). The letters (a,b) next to the symbol * indicate the groups that are statistically different after *Post Hoc* Bonferroni correction.

Biological Width	Immediate Implants (n = 40)		Delayed Implants (n = 40)		Comparisons by	
					Student T; *p*-Value	
	Buccal	Lingual	Buccal	Lingual	Buccal	Lingual
	Mean (sd)	Mean (sd)	Mean (sd)	Mean (sd)		
Strauman BL	3.8(0.7)	3.1(1.7) *	3.2(0.6)	1.8(0.2) *^,a^	T = 1.6 *p* = 0.13	T = 2.2 *p* = 0.04
IPX-Std	4.4(2.3)	3.9(1.0) *	4.0(0.7)	3.0(0.5) *^,b^	T = 0.4 *p* = 0.68	T = 2.1 *p* = 0.05
IPX-Half	4.6(2.4)	4.4(2.3)	4.2(1.1)	2.9(0.5)	T = 0.4 *p* = 0.72	T = 1.8 *p* = 0.13
IPX-Full	4.5(1.8)	3.9(1.1) *	4.1(0.3)	2.7(0.6) *	T = 0.6 *p* = 0.56	T = 2.5 *p* = 0.03
IPX-Control	5.1(1.1)	4.7(1.3) *	4.3(0.6)	2.9(1.1) *^,b^	T = 1.5 *p* = 0.18	T = 2.8 *p* = 0.02
Comparisons by ANOVA: *p*-value	F = 0.4 *p* = 0.79	F = 1.1 *p* = 0.40	F = 2.1 *p* = 0.11	F = 3.3 *p* = 0.02		

**Table 3 materials-13-00802-t003:** Comparison of the epithelial width (the distance between PM and aJE) between deferred and immediate implants in both the buccal and lingual sides (mm) among the implant subgroups. * Statistically significant inter-group comparisons according to ANOVA test (*p* < 0.05).

Epithelial Width	Immediate Implants (n = 40)		Delayed Implants (n = 40)		Comparisons by	
					Student T; *p*-Value	
	Buccal	Lingual	Buccal	Lingual	Buccal	Lingual
	Mean (sd)	Mean (sd)	Mean (sd)	Mean (sd)		
Strauman BL	2.0(0.6)	2.1(1.7)	2.2(0.5)	0.9(0.3)	T = 0.5 *p* = 0.65	T = 1.8 *p* = 0.11
IPX-Std	2.6(1.3)	2.0(1.1)	2.1(0.9)	1.4(0.5)	T = 0.9 *p* = 0.39	T = 1.3 *p* = 0.22
IPX-Half	2.7(1.3)	2.1(1.1)	2.2(0.5)	1.2(0.7)	T = 1.1 *p* = 0.30	T = 1.9 *p* = 0.08
IPX-Full	2.9(1.0) *	1.9(0.6)	2.1(0.5) *	1.3(0.7)	T = 2.1 *p* = 0.05	T = 1.8 *p* = 0.10
IPX-Control	2.4(0.9)	1.8(0.5)	2.3(0.6)	1.3(0.5)	T = 0.2 *p* = 0.82	T = 1.9 *p* = 0.08
Comparisons by ANOVA;	F = 0.7 *p* = 0.60	F = 0.1 *p* = 0.97	F = 0.1 *p* = 0.97	F = 0.8 *p* = 0.53		
*p*-value						

**Table 4 materials-13-00802-t004:** Comparison of the connective width (the distance between aJE and B) between delayed and immediate implants in both the buccal and lingual sides (mm) among the implant subgroups. * Statistically significant inter-group comparisons according to ANOVA test (*p* < 0.05). The letters (a,b) next to the symbol * indicate the groups that are statistically different after post hoc Bonferroni correction.

Connective Width	Immediate Implants (n = 40)		Delayed Implants (n = 40)		Comparisons by	
					Student T; *p*-Value	
	Buccal	Lingual	Buccal	Lingual	Buccal	Lingual
	Mean (sd)	Mean (sd)	Mean (sd)	Mean (sd)		
Strauman BL	1.8(0.6) *	1.1(0.5) ^a^	1.1(0.5) *^,a^	1.0(0.2)	T = 2.5 *p* = 0.03	T = 0.3 *p* = 0.75
IPX-Std	2.1(1.5)	2.0(1.2)	1.9(0.5)	1.6(0.5)	T = 0.3 *p* = 0.75	T = 0.7 *p* = 0.51
IPX-Half	2.2(1.4)	2.3(1.5)	2.0(1.0) ^b^	1.6(0.5)	T = 0.3 *p* = 0.75	T = 1.1 *p* = 0.30
IPX-Full	1.6(1.0)	2.0(0.6) *	2.0(0.6)	1.5(0.2) *	T = 0.8 *p* = 0.45	T = 2.2 *p* = 0.05
IPX-Control	2.7(1.2)	3.0(1.3) *^,b^	2.0(0.2)	1.7(0.8) *	T = 1.6 *p* = 0.16	T = 2.2 *p* = 0.05
Comparisons by ANOVA; *p*-value	F = 0.8 *p* = 0.54	F = 2.7 *p* = 0.05	F = 2.6 *p* = 0.05	F = 2.0 *p* = 0.13		

**Table 5 materials-13-00802-t005:** Comparison of the marginal bone loss (the distance between IS and B) between delayed and immediate implants in both the buccal and lingual sides (mm) among the implant subgroups. * Statistically significant inter-group comparisons according to ANOVA test (*p* < 0.05).

Marginal Bone Loss	Immediate Implants (n = 40)		Delayed Implants (n = 40)		Comparisons by	
					Student T; *p*-Value	
	Buccal	Lingual	Buccal	Lingual	Buccal	Lingual
	Mean (sd)	Mean (sd)	Mean (sd)	Mean (sd)		
Strauman BL	1.3(0.8) *	0.6(0.3)	0.6(0.4) *	0.4(0.3)	T = 2.5 *p* = 0.03	T = 1.6 *p* = 0.13
IPX-Std	0.3(0.9)	0.3(0.4)	0.2(0.3)	0.4(0.5)	T = 0.4 *p* = 0.70	T = 0.2 *p* = 0.83
IPX-Half	0.8(0.6)	0.5(0.4) *	0.7(1.3)	0.1(0.3) *	T = 0.2 *p* = 0.81	T = 2.3 *p* = 0.04
IPX-Full	1.0(0.9)	0.3(0.4)	0.3(0.6)	0.2(0.4)	T = 1.8 *p* = 0.10	T = 0.4 *p* = 0.68
IPX-Control	0.8(1.4)	0.9(0.8)	1.6(2.2)	0.9(1.1)	T = 0.9 *p* = 0.31	T = 0.0 *p* = 0.99
Comparisons by ANOVA; *p*-value	F = 1.2 *p* = 0.35	F = 2.0 *p* = 0.13	F = 1.9 *p* = 0.14	F = 2.0 *p* = 0.12		

**Table 6 materials-13-00802-t006:** Comparison of the BIC (bone-to-implant-contact) between delayed and immediate implants in both the buccal and lingual sides (mm) among the implant subgroups. The letters (a,b) indicate the groups that are statistically different after post hoc Bonferroni correction.

BIC	Immediate Implants (n = 40)	Delayed Implants (n = 40)	Comparisons by
			Student T; *p*-value
	Mean (sd)	Mean (sd)	
Strauman BL	54.8(22.9)	64.6(19.1)	T = 0.9 *p* = 0.37
IPX-Std	62.3(16.3)	68.6(21.3)	T = 0.7 *p* = 0.52
IPX-Half	72.5(12.8) ^a^	72.7(9.8)	T = 0.0 *p* = 0.97
IPX-Full	63.9(19.6)	74.2(15.6)	T = 1.2 *p* = 0.26
IPX-Control	43.2(15.2) ^b^	52.9(21.3)	T = 1.1 *p* = 0.31
Comparisons by ANOVA; *p*-value	F = 3.1 *p* = 0.03	F = 1.8 *p* = 0.15	

**Table 7 materials-13-00802-t007:** Linear regression model (backward stepwise) for predicting the average biological width and the average marginal bone loss after including all the potential predictors (time of implant placement and grade of surface treatment). * For these analyses, the mechanized implants placed on the M_1_ were excluded to assess the true effect of the platform configuration. † F = 6.97: df:2; *p*-value = 0.001. R^2^ = 0.29. ‡ F = 3.17: df:2; *p*-value = 0.031. R^2^ = 0.14.

Dependent Variable/Parameters	Standardized β	*p*-Value	β	CI-95% β	
				Lower	Upper
**Average Biological Width †**					
Constant (mm)		<0.001	3.5	2.9	4.1
Time of placement (immediate as reference)	−0.43	0.001	−1.0	−0.5	−1.6
Grade of surface treatment (none/half/full)	1.0	0.007	1.1	0.3	1.9
**Average Marginal Bone Loss ‡**					
Constant (mm)		<0.001	0.8	0.6	1.1
Time of placement (immediate as reference)	−0.24	0.04	−0.3	−0.1	−0.5
Grade of surface treatment (none/half/full)	−0.83	0.02	−0.4	−0.1	−0.8

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
