# Peer review of "Effect of Rough Surface Platforms on the Mucosal Attachment and the Marginal Bone Loss of Implants: A Dog Study"

_materials, 2020, doi:10.3390/ma13030802_

Round 1
Reviewer 1 Report
This manuscript aims to describe the impact of surface roughness of peri-implant on soft tissue healing and bone loss placed immediately or delayed post tooth extraction. The aim of the study is clear and the data presented is of a quality that can be properly analysed to conclude the usefulness for clinical applications. Some clarification is however needed to better answer the research question.
Major comments:
Implant characteristics The details of the implants is limited. As the main purpose of this study is to evaluate the impact of surface roughness, this parameter should be disclosed. It is furthermore hard to follow the text describing the implant details, it does not look very good listing bullet-point. Please rearrange the information into a table which is easy to overview, including additional data missing. Information about how and to what extend these implants are applied today would be useful. Figure 1 needs additional information provided in the figure text to disclose the differences. It would be useful to have a high magnification image of the actual surface to help visualise the difference. Also add numbering to this figure to match with later figures. The authors need to add more discussion about why a difference is expected is expected between the samples. What is the rationale behind changing from std to half to full? It is not made clear why the IPX-xontrol implant is always inserted at the most distal site (M1). The nomenclature is confusing, the text refers to one sample as SLA but it is referred to as Straumann BL in the figure. Results It is not clear if the samples in Figure 2 and 3 is from the same patient or if they are best of photos. A discussion about the success of the animal model design should be added and how this compares to current literature. Conclusions The authors state that “The major finding in the present study is the fact that both the height of the supracrestal connective tissue and the length of the junctional epithelium were significantly greater at implants immediately placed in extraction sockets compared to implants placed in healed sites”. Yet, this finding is not included in the conclusion section, instead the authors only focus on the impact of the surface roughness, stating it is inconclusive. Details about why the study is inconclusive should be discussed and what adjustments should be done to make it better. It is furthermore contradicting to read that the rough platform may be beneficial when the study was deemed inconclusive.
Author Response
Dear Reviewer,
Thank you for your time and comments that certainly improve the quality and clarity of our work.
Following your recommendations, the following modifications were completed:
Reviewer:
Implant characteristics The details of the implants is limited. As the main purpose of this study is to evaluate the impact of surface roughness, this parameter should be disclosed.
Authors:
We have added roughness values as recommended (Lines 110-125) and table 1.
Reviewer:
Please rearrange the information into a table which is easy to overview, including additional data missing. Information about how and to what extend these implants are applied today would be useful.
Authors:
A table (Table 1) has been included summarizing all the information and providing additional data as suggested.
Reviewer:
Figure 1 needs additional information provided in the figure text to disclose the differences.
Authors:
The Figure text has been modified with extensive information about platform differences.
Reviewer:
It would be useful to have a high magnification image of the actual surface to help visualize the difference.
Authors:
A new figure (Figure 2) with high magnification images has been added.
Reviewer:
The authors need to add more discussion about why a difference is expected between the samples.
Authors:
New paragraphs have been redacted into the discussion part following your recommendations.
Reviewer:
What is the rationale behind changing from std to half to full?
Authors:
We hardly believe that platform features affect bone response. By introducing the variable of the different surface treatments in the implant platform, we can observe the bone response behavior and compare the differences between the parameters evaluated in this study. It is not expected the same response between a rough surface and a machined one.
Reviewer:
It is not made clear why the IPX-control implant is always inserted at the most distal site (M1).
Authors:
There is some bibliography that suggests that due to the anatomical differences, M1 is the ideal site for placing controls.
Vignoletti F., Sanz-Esporrin J., Sanz-Martin I., Nuñez J., Luengo F, Sanz M. Ridge alterations after implant placement in fresh extraction sockets or in healed crest: An experimental in vivo investigation. Clin Oral Implants Res. 2019 Apr; 30(4):353-363.
Reviewer:
The nomenclature is confusing, the text refers to one sample as SLA but it is referred to as Straumann BL in the figure
Authors:
SLA nomenclature has been changed for Straumann BL, as suggested. In fact, SLA is the name of the surface treatment.
Reviewer:
It is not clear if the samples in Figure 2 and 3 is from the same patient or if they are best of photos.
Authors:
Figure 2 (now Figure 3) refers to Histological images of “immediate implants” and Figure 3 (now Figure 4) to Histological images of “delayed implants”. Indeed, they are not images of the same patient but randomly chosen for each type of implant (immediate or delayed).
Reviewer:
Conclusions The authors state that “The major finding in the present study is the fact that both the height of the supracrestal connective tissue and the length of the junctional epithelium were significantly greater at implants immediately placed in extraction sockets compared to implants placed in healed sites”. Yet, this finding is not included in the conclusion section, instead the authors only focus on the impact of the surface roughness, stating it is inconclusive. Details about why the study is inconclusive should be discussed and what adjustments should be done to make it better. It is furthermore contradicting to read that the rough platform may be beneficial when the study was deemed inconclusive.
Authors:
In our initial conclusions, we just concluded about the influence of the topographical platform design as inconclusive. You are right, and we didn’t conclude anything about a second topic, which is the “timing for placing the implant,” that is: immediate vs. delayed. We have added a paragraph in “conclusions” about this topic which confirm the statement that “The major finding in the present study is the fact that both the height of the supracrestal connective tissue and the length of the junctional epithelium were significantly greater at implants immediately placed in extraction sockets compared to implants placed in healed sites.”
Reviewer 2 Report
Dear Authors,
thank you very much for your paper. It speaks about the role of the platform surface roughness on soft tissue attachment and marginal bone loss. A split mouth design was used in this study where on one side post extractions implants were inserted meanwhile on the other side implants were inserted on healed sites after a healing period of 12 weeks. It is a very interesting subject on which nowadays the worldwide research in the Implantology field is being focused. However, I have some queries. Please find attached my comments:
Affilations: typos are present. please put them on the journal style and be careful that sometime you have put ";" before the email address and others "."
Abstract: please rearrange better the sentences from 27-29 for a better understanding of the procedures. It is not immediate (and it should be in the abstract) that the implants were inserted in post extracted sockets by one side and in healed ones in two different time points.
line 30-31: you stated that" Implants were inserted 1 mm subcrestal to the buccal bony plate and were connected to abutments." Why you chose to place them subcrestal on the buccal bone plate and how did you get it? How was it possible to have implants placed subcrestal only on buccal plate and not on the other sides, too?
line 32: could you please explain with a sentences how the SST was measured?
Introduction:
line 48-49 express the same concept of line 50-51. The latter has no reference. Could you please rearrange or cancel one of them? The introduction sections is too skimpy. Could you please enrich it also with results of other studies on the marginal bone loss and on soft tissue marginal seal to better understand the necessity of you study?Please put the lates references regarding.
Materials and methods: Please could you explain why the study was approved by the Ethical Committee of the University of Santiago of Compostela? No one of the authors seems to have an affiliation on this University.Is it for administrative reasons? It looks so strange...Line 78-79...you mean that it was a blinded study? if yes please specify it in the text.
line 94: please specify the manufacture, address, etc info for the multi unit abutments. line 96: the multi unit abutments in this case are the same of the line 94 if no please specify why you chose others? if the same please specify it in the text. It is important to better understand the study as not only the abutments surface but also the geometry of them may influence the soft tissues. Line 118, you said that the implants were randomly assigned on the location but why you state here that the IPX control implants have been inserted to the most distal site? Why you did not included also the control implant on the randomization chart? The author chose 5 different implants, which have not only different surfaces but also different threads designs (see bone level implant with IPX...) and different dimensions 3,3 VERSUS 3.5 respectively for bl and IPX.WHY? line 168: here you speak about BIC evaluation but you did not mention it in the abstract. Regarding the marginal bone loss no initial time point was evaluated. Who says that the differences between the groups were not due to a different initial bone level? This is a lack of the methodology form my point of view. Please specify why you did not evaluated, probably with a radiography immediately after implant placement and one immediately before block section ?
Discussion
generally it is well written. Please revise the English language and write down the limitations of the present study.
Author Response
Dear Reviewer,
Thank you for your time and comments that certainly improve the quality and clarity of our work.
Following your recommendations, the following modifications were completed:
REVIEWER:
Affilations: typos are present. please put them on the journal style and be careful that sometime you have put ";" before the email address and others "."
AUTHORS:
Pertinent changes have been done.
REVIEWER:
Abstract: please rearrange better the sentences from 27-29 for a better understanding of the procedures. It is not immediate (and it should be in the abstract) that the implants were inserted in post extracted sockets by one side and in healed ones in two different time points.
AUTHORS:
In line 33 “sockets” have been changed for “post-extracted-socket” to clarify this point.
REVIEWER:
line 30-31: you stated that" Implants were inserted 1 mm subcrestal to the buccal bony plate and were connected to abutments." Why you chose to place them subcrestal on the buccal bone plate and how did you get it? How was it possible to have implants placed subcrestal only on buccal plate and not on the other sides, too?
AUTHORS:
The reference to measure 1mm was the buccal bone plate, but the whole platform was placed subcrestally.
REVIEWER:
line 32: could you please explain with a sentences how the SST was measured?
AUTHORS:
The supracrestal soft tissue height is the distance between PM and B as explained in material and methods, line 209 and Figure 5. It was measured in the histological preparations. It is not explained in the abstract due to the few words allowed by the journal (just 200).
REVIEWER:
Introduction:
line 48-49 express the same concept of line 50-51. The latter has no reference. Could you please rearrange or cancel one of them? The introduction sections is too skimpy. Could you please enrich it also with results of other studies on the marginal bone loss and on soft tissue marginal seal to better understand the necessity of you study?Please put the lates references regarding.
AUTHORS:
In the first paragraph (lines 47-49) we introduce the healing dynamics in both hard and soft tissues, while in the second paragraph (lines 50-51) we focus on hard tissue trying to see differences between immediate and delayed placement, so we think they are supplementary information and one should not exclude the other.
You have all the reasons that the introduction is too short, but we tried to follow the journal recommendations with a brief introduction. In any case, we agree to extend that section following your recommendations.
REVIEWER:
Materials and methods: Please could you explain why the study was approved by the Ethical Committee of the University of Santiago of Compostela? No one of the authors seems to have an affiliation on this University.Is it for administrative reasons? It looks so strange...Line 78-79...you mean that it was a blinded study? if yes please specify it in the text.
AUTHORS:
Indeed, the study was approved by the Ethical Committee of the University of Santiago of Compostela as the experimental phase was developed in its facilities,as well as histological measurements, by the hand of Dr. Fernando Muñoz, as you can see in “Acknowledgments”.
It wasn´t a blinded study.
REVIEWER:
e 94: please specify the manufacture, address, etc info for the multi unit abutments. line 96: the multi unit abutments in this case are the same of the line 94 if no please specify why you chose others? if the same please specify it in the text.
AUTHORS:
Each Multi-Unit abutment was by the same manufacturer as its corresponding implant. It has been specified in the text; and manufacturer, address, etc has been added.
REVIEWER:
It is important to better understand the study as not only the abutments surface but also the geometry of them may influence the soft tissues. Line 118, you said that the implants were randomly assigned on the location but why you state here that the IPX control implants have been inserted to the most distal site? Why you did not included also the control implant on the randomization chart?
AUTHORS:
Following the recommendations of other authors, we inserted the control group into the M1. They suggest that due to the anatomical differences, M1 is the ideal site for placing controls.
Vignoletti F., Sanz-Esporrin J., Sanz-Martin I., Nuñez J., Luengo F, Sanz M. Ridge alterations after implant placement in fresh extraction sockets or in healed crest: An experimental in vivo investigation. Clin Oral Implants Res. 2019 Apr; 30(4):353-363.
REVIEWER:
The author chose 5 different implants, which have not only different surfaces but also different threads designs (see bone level implant with IPX...) and different dimensions 3,3 VERSUS 3.5 respectively for bl and IPX.WHY?
AUTHORS:
We decided to use Straumann BL because it’s an implant widely used in previous studies and has a non-treated platform, si it could work perfectly as the reference control compared with the four variations of the experimental implant (IPX with four different surface treatments). Being different implant systems from different manufacturers, the thread designs and dimensions are slightly different. A new paragraph has been added in “discussion” (lines 333 to 340), clarifying this point.
REVIEWER:
Regarding the marginal bone loss no initial time point was evaluated. Who says that the differences between the groups were not due to a different initial bone level? This is a lack of the methodology form my point of view. Please specify why you did not evaluated, probably with a radiography immediately after implant placement and one immediately before block section ?
AUTHORS:
We did not directly measure the initial bone level, but indirectly, during the surgery, all implants were placed exactly 1 mm from the buccal plate, so our initial time point was the implant platform location plus 1 mm. The precision measuring the platform location with a probe at the time of placing the implant could be even more precise than measuring radiography.
REVIEWER:
Discussion
generally it is well written. Please revise the English language and write down the limitations of the present study.
AUTHORS:
a specialized program has revised language style and limitations added.
Reviewer 3 Report
It is a good study, but does not give readers more interpretation on the results, which is the main issue in this study.
The topic is the effect of rough surface platforms, but less discussion on the roughness is represented in the Discussion section.
Author Response
Dear Reviewer,
Thank you for your time and comments that certainly improve the quality and clarity of our work.
Following your recommendations, a wider interpretation of the results has been added to the discussion (lines 353 to 371). Also, the roughness topic has been addressed in a more specific way.
Reviewer 4 Report
The work presents an accurate and very well exposed scientific protocol; accurate experimental methodology; very interesting work regarding the scientific relevance of the topic. Only the discussion part could have been more developed by comparing the results obtained with those in the literature.
Author Response
The work presents an accurate and very well exposed scientific protocol; accurate experimental methodology; very interesting work regarding the scientific relevance of the topic. Only the discussion part could have been more developed by comparing the results obtained with those in the literature.
Dear Reviewer,
Thank you for your review and comments that certainly improve the quality and clarity of our work.
Following your recommendations, a wider interpretation of the results has been added to the discussion (lines 353 to 371).
Round 2
Reviewer 3 Report
Good enough for publication.